# DualBind: A Dual-Loss Framework for Protein-Ligand Binding Affinity Prediction

Meng Liu [1]    Saee Gopal Paliwal [1]

## Abstract

Accurate prediction of protein-ligand binding affinities is crucial for drug development. Recent advances in machine learning show promising results on this task. However, these methods typically rely heavily on labeled data, which can be scarce or unreliable, or they rely on assumptions like Boltzmann-distributed data that may not hold true in practice. Here, we present DualBind, a novel framework that integrates supervised mean squared error (MSE) with unsupervised denoising score matching (DSM) to accurately learn the binding energy function. DualBind not only addresses the limitations of DSM-only models by providing more accurate absolute affinity predictions but also improves generalizability and reduces reliance on labeled data compared to MSE-only models. Our experimental results demonstrate that DualBind excels in predicting binding affinities and can effectively utilize both labeled and unlabeled data to enhance performance.

## 1. Introduction

Accurate prediction of protein-ligand binding energies is a fundamental task in drug discovery workflow, as it directly influences the prioritization and optimization of potential drug candidates by evaluating their interaction strength with the target. This task presents significant challenges due to the complex nature of protein-ligand interactions and the intricate potential energy landscape they create. Traditionally, various scoring functions designed by experts, such as those in Glide (Friesner et al., 2004) and AutoDock Vina (Trott & Olson, 2010; Eberhardt et al., 2021), have been employed to estimate binding affinities. In recent years, data-driven machine learning methods have also emerged, offering new avenues for enhancing prediction accuracy through flexible and learnable neural networks (Jiménez et al., 2018; Öztürk et al., 2018; Chen et al., 2020; Li et al., 2020; Nguyen et al., 2021; Lu et al., 2022; Jiang et al., 2021; Moon et al., 2022; Zhang et al., 2023a; Jin et al., 2023b; Lin et al., 2024; Zhang et al., 2023b).

The most straightforward approach in machine learning is to train a model that minimizes prediction errors directly, requiring reliable binding affinity labels for protein-ligand complexes. Notably, models relying solely on a supervised loss might overfit to noisy or sparse data, reducing their robustness and generalizability. Recently, DSMBind (Jin et al., 2023b;a) adopted a generative modeling strategy by training an energy-based model (EBM) with a denoising score matching (DSM) objective. This DSM-driven approach aimed to maximize the log-likelihood of crystal structures within the training set. While this approach did not yield direct comparisons to actual binding affinity values, it suggests that the learned energy function correlates with experimental binding energies. DSMBind only needs the protein-ligand complex structures for training, without requiring binding affinity labels.

In this work, we first identify and analyze an unrealistic assumption in DSM-trained model; that is, the distribution of training data follows a Boltzmann distribution. Then we introduce DualBind, a simple and novel framework which synergistically combines the supervised mean squared error (MSE) objective with the denoising score matching (DSM) objective to learn the energy function of protein-ligand interactions. Compared to models relying solely on MSE or DSM, DualBind offers several notable advantages. DSM-only models often struggle with the Boltzmann distribution assumption, that does not reliably hold within the training set. In contrast, DualBind aligns closely with actual training affinity data, which significantly enhances the performance of energy predictions. Crucially, this alignment allows DualBind to predict absolute affinity values, rather than merely comparative values. Compared to MSE-only methods that depend heavily on labeled data and may suffer from overfitting, DualBind achieves better generalization by incorporating the DSM objective. In addition, DualBind is not strictly dependent on the availability of labeled data. It can effectively utilize additional unlabeled data by learning the broader energy distribution, which is especially valuable

[1]NVIDIA. Correspondence to: Meng Liu <menliu@nvidia.com>, Saee Gopal Paliwal <saeep@nvidia.com>.

*Accepted at the 1st Machine Learning for Life and Material Sciences Workshop at ICML 2024*. Copyright 2024 by the author(s).

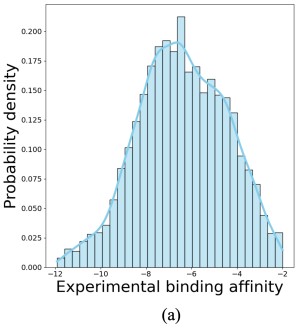 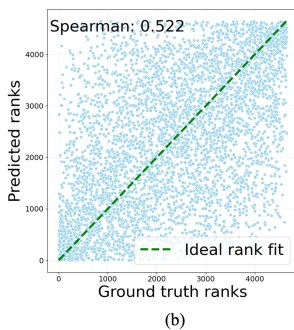

(a)          (b)

*Figure 1.* (a) **Distribution of binding affinity in the PDBbind v2020 refined dataset.** It departs from the Boltzmann distribution. (b) **Rank fit of a DSM-only model on training complexes.**

in scenarios where experimental data are scarce.

To validate the benefits of DualBind, we conducted extensive experiments on the task of protein-ligand binding affinity prediction. The results consistently show that DualBind outperforms both DSM-only and MSE-only models across a variety of metrics. Furthermore, our empirical evidence demonstrates that DualBind's capability to simultaneously train on both labeled and unlabeled data significantly enhances model performance.

## 2. Method

In this section, we first point out the unrealistic nature of the underlying Boltzmann distribution assumption typically employed in DSM-only Models in Section 2.1. Then, in Section 2.2, we present our proposed DualBind approach and describe its unique advantages.

### 2.1. Boltzmann Distribution Assumption in DSM-only Models

Energy-based models (EBMs) are a class of models that represent complex systems through an energy function. These models assign a scalar energy value to each configuration of the system (LeCun et al., 2006; Song & Kingma, 2021). A denoising score matching (DSM) objective has been widely used to train EBMs (Vincent, 2011; Song & Ermon, 2019; Ho et al., 2020; Song et al., 2021; Jin et al., 2023a). The efficacy of the DSM objective, which focuses on accurately learning the energy function by maximizing the likelihood of the training data, inherently relies on the assumption that the distribution of training samples aligns with the Boltzmann distribution. Specifically, in the context of protein-ligand binding, this assumption implies that the probability $p(C)$ of a complex $C$ being observed is proportional to $\exp\left(-E(C)\right)$, where $E(C)$ denotes its corresponding binding energy. Note that a lower binding energy indicates a stronger interaction between a ligand and a protein.

However, the actual distribution of protein-ligand complexes in experimental datasets often diverges from this assumption due to experimental biases, selective data reporting, and other confounding factors (Li et al.; Hernández-Garrido & Sánchez-Cruz, 2023). These discrepancies typically result in a dataset that does not exhibit the expected exponential decay of energy states. To validate this observation, we analyze the distribution of binding affinity in the PDB-bind v2020 refined dataset, a widely used training dataset where the affinity data are collected from the scientific literature (Su et al., 2018). As illustrated in Figure 1 (a), the binding affinity distribution significantly departs from the expected Boltzmann distribution. Consequently, although the learned EBM can effectively assign local minima to observed protein-ligand complexes, as demonstrated in Jin et al. (2023b), we conjecture it struggles to accurately differentiate between their relative binding affinities.

To verify this, we reproduce an EBM learned by a DSM objective based on Gaussian noise. The learned model achieves a Pearson correlation coefficient of 64.2 on the CASF-2016 test benchmark (Su et al., 2018), which is comparable to the performance of DSMBind (Jin et al., 2023b). To evaluate the model's capability in learning relative binding affinities from the training dataset, we analyzed the rank fit results *on the training set* using the learned EBM model, as illustrated in Figure 1 (b). The plot shows the predicted versus ground truth ranks of training protein-ligand complexes. Ideally, these ranks should align closely along the dashed green line, which represents perfect correlation. However, there is a significant scatter with a Spearman correlation coefficient of only 0.522. This divergence indicates that DSM-only models struggle to accurately rank the binding affinities of protein-ligand complexes. This finding supports our hypothesis that the DSM's effectiveness is compromised by its dependence on a theoretical Boltzmann distribution that does not hold in the widely used protein-ligand complex dataset.

### 2.2. DualBind: A Dual-Loss Approach

Here, we propose DualBind, which employs a dual-loss framework to effectively learn the energy function. This integration is not only simple and straightforward but also provides crucial advantages over models that rely solely on DSM or MSE loss. In the following, we first describe the overall dual-loss framework, then introduce the architecture of the energy function model, and finally discuss the significant benefits of DualBind compared to DSM-only and MSE-only models.

**The Dual-Loss Framework.** We represent a protein-ligand complex structure as $C = (\boldsymbol{A}, \boldsymbol{X})$, where $\boldsymbol{A} \in \mathbb{R}^{n \times d}$ denotes the features of the $n$ atoms in the complex, including both protein and ligand atoms, and $\boldsymbol{X} \in \mathbb{R}^{n \times 3}$ represents

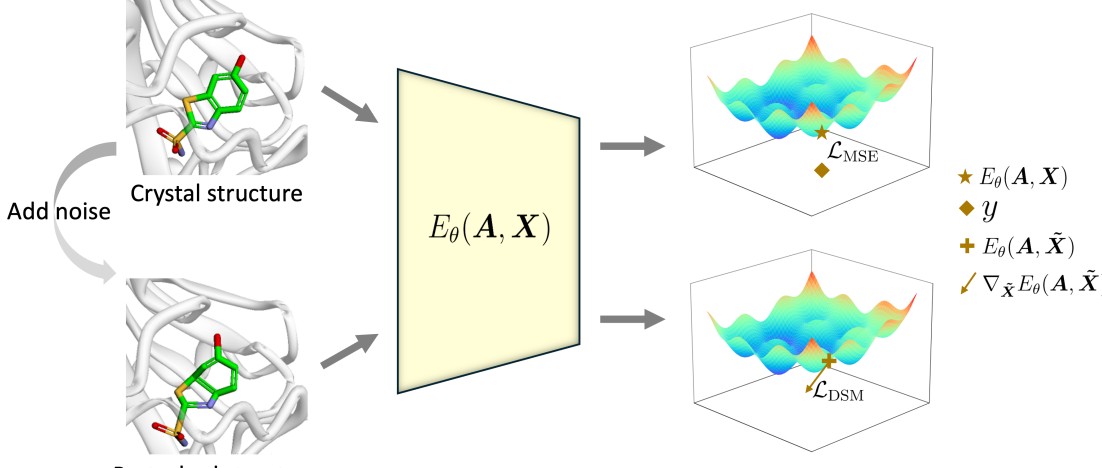

*Figure 2.* **An illustration of the DualBind methodology.** DualBind employs a dual-loss framework that combines the MSE loss $\mathcal{L}_{\text{MSE}}$ and the DSM loss $\mathcal{L}_{\text{DSM}}$. Specifically, $\mathcal{L}_{\text{MSE}}$ anchors the predicted binding affinity of the crystal structure to its experimentally determined binding affinity. Concurrently, $\mathcal{L}_{\text{DSM}}$ shapes the gradient at the perturbed structure. Details are described in Section 2.

the coordinates of these atoms. We define a binding affinity prediction model as $E_\theta(\boldsymbol{A}, \boldsymbol{X}) : \mathbb{R}^{n \times d} \times \mathbb{R}^{n \times 3} \to \mathbb{R}$, which produces a scalar energy value for any given complex. $\theta$ denotes the learnable parameters in the model.

Intuitively, our dual-loss framework combines the DSM loss $\mathcal{L}_{\text{DSM}}$, which learns the energy landscape by shaping the gradient of the energy function, with the MSE loss $\mathcal{L}_{\text{MSE}}$, which directly ties the predictions to known binding affinity values. The overview of our DualBind approach is illustrated in Figure 2.

*MSE Loss.* As shown in the upper branch of Figure 2, the use of the MSE loss is straightforward. We feed a complex crystal structure $(\boldsymbol{A}, \boldsymbol{X})$ into the model and then compute the error between the predicted and the ground truth binding affinity. Formally, for one data sample,

$$\mathcal{L}_{\text{MSE}} = \left( E_\theta(\boldsymbol{A}, \boldsymbol{X}) - y \right)^2, \qquad (1)$$

where $y$ denotes the the ground truth binding affinity. Intuitively, this MSE loss encourages that the points, which represent observed crystal structures within the energy landscape, are anchored to experimental binding affinities, maintaining their alignment with empirical observations.

*DSM Loss.* Training neural networks with denoising is a well-established technique aimed at enhancing the robustness and generalization capabilities of models (Bishop, 1995; Vincent et al., 2008; Kong et al., 2020; Godwin et al., 2021). More specifically, the denoising score matching technique has been employed to pretrain models for 3D molecular tasks, demonstrating that such objective not only simulates learning a molecular force field but also significantly boosts performance across various downstream tasks (Zaidi et al., 2022). Furthermore, within the framework of energy-

based models (EBMs), the DSM objective can be seen as a way of modeling the likelihood of the data (Song & Ermon, 2019; Ho et al., 2020; Song et al., 2021; Jin et al., 2023a). By training an EBM to denoise perturbed data, we implicitly maximize the likelihood of observing the original, unperturbed data under the modeled distribution. Thus, in the context of protein-ligand binding energy modeling, the DSM objective ensures that a crystal structure is guided toward the local minima of the learned energy landscape.

Given such generalization capability and ability to accurately shape local minima for crystal structures, we incorporate the DSM objective into our DualBind model. Following previous DSM-based studies (Zaidi et al., 2022; Jin et al., 2023b), given a crystal struture $(\boldsymbol{A}, \boldsymbol{X})$ from the training dataset, we first perturb it by adding Gaussian noise specifically to the ligand atoms' coordinates. Formally, for each ligand atom coordinate $\boldsymbol{x}_i$, the perturbed coordinate $\tilde{\boldsymbol{x}}_i$ can be obtained by

$$\tilde{\boldsymbol{x}}_i = \boldsymbol{x}_i + \sigma \boldsymbol{\epsilon}_i, \quad \text{where } \boldsymbol{\epsilon}_i \sim \mathcal{N}(0, \boldsymbol{I}_3). \qquad (2)$$

$\sigma$ is a hyperparameter for tuning the noise scale. Note that Gaussian noise is added only to the coordinates of the ligand atoms in $\boldsymbol{X}$. However, for simplicity in notation, we denote the resulting matrix, which includes both perturbed ligand atom coordinates and unperturbed protein atom coordinates, as $\tilde{\boldsymbol{X}}$.

As illustrated in the lower branch in Figure 2, we feed the perturbed complex structure into the model and obtain its predicted energy $E_\theta(\boldsymbol{A}, \tilde{\boldsymbol{X}})$. Then the DSM loss matches the score of the model distribution $p_\theta$ at the perturbed data $\tilde{\boldsymbol{X}}$ with the score of the distribution $q$ where $\tilde{\boldsymbol{X}}$ is sampled. Score is defined as the gradient of the log-probability *w.r.t.*

$\tilde{X}$. Formally,

$$\mathcal{L}_{\text{DSM}} = \mathbb{E}_{q(\tilde{X})} \left[ \left\| \nabla_{\tilde{X}} \log p_\theta(A, \tilde{X}) - \nabla_{\tilde{X}} \log q(\tilde{X}) \right\|^2 \right]. \tag{3}$$

As shown by Vincent (2011), this is equivalent to

$$\mathcal{L}_{\text{DSM}} = \mathbb{E}_{q(\tilde{X}|X)p_{\text{data}}(X)} \left[ \left\| \nabla_{\tilde{X}} \log p_\theta(A, \tilde{X}) \right. \right.$$
$$\left. \left. - \nabla_{\tilde{X}} \log q(\tilde{X}|X) \right\|^2 \right]. \tag{4}$$

The distribution given by $E_\theta(A, \tilde{X})$ is defined as $p_\theta(A, \tilde{X}) = \frac{\exp(-E_\theta(A, \tilde{X}))}{Z_\theta}$, where $Z_\theta$ is the normalizing constant, which is typically intractable but a constant *w.r.t.* $\tilde{X}$ (LeCun et al., 2006; Song & Kingma, 2021). Therefore, the first term in Eq. (4) can be computed by

$$\nabla_{\tilde{X}} \log p_\theta(A, \tilde{X}) = -\nabla_{\tilde{X}} E_\theta(A, \tilde{X}) - \nabla_{\tilde{X}} \log Z_\theta$$
$$= -\nabla_{\tilde{X}} E_\theta(A, \tilde{X}). \tag{5}$$

The second term in Eq. (4) can be easily computed as $\nabla_{\tilde{X}} \log q(\tilde{X}|X) = -\frac{(\tilde{X}-X)}{\sigma^2}$ since $q(\tilde{X}|X)$ is a Gaussian distribution. Hence,

$$\mathcal{L}_{\text{DSM}} = \mathbb{E}_{q(\tilde{X}|X)p_{\text{data}}(X)} \left[ \left\| \nabla_{\tilde{X}} E_\theta(A, \tilde{X}) - \frac{(\tilde{X}-X)}{\sigma^2} \right\|^2 \right]. \tag{6}$$

Intuitively, by applying this DSM loss, the model learns to shape its energy landscape such that the energy valleys (*a.k.a.*, local minima) align with the original unperturbed crystal structures.

The total training loss is defined as the weighted summation of the MSE loss and the DSM loss, *i.e.*,

$$\mathcal{L} = \mathcal{L}_{\text{MSE}} + \lambda \mathcal{L}_{\text{DSM}}, \tag{7}$$

where $\lambda$ is a hyperparameter that balances the influence of two losses.

**The Architecture of $E_\theta(A, X)$.** To facilitate a meaningful comparison with DSMBind, we adopt the architecture for $E_\theta(A, X)$ as used in in DSMBind. It incorporates an SE(3)-invariant model, built on a frame averaging neural network (Puny et al., 2021), to obtain the atom representations. Energy predictions are then made by considering pairwise atom interactions within a predefined distance threshold. For an in-depth understanding of this architecture, we direct readers to Jin et al. (2023b). It is important to note that our dual-loss framework is model-agnostic. This flexibility ensures that our framework can be easily adapted across different modeling approaches for $E_\theta(A, X)$.

**Advantages of DualBind.** DualBind offer substantial benefits over models that uses only DSM or MSE.

- Compared to DSM-only models such as DSMBind, DualBind's integration with actual training affinity data significantly boosts the prediction performance.

- More importantly, this alignment enables the prediction of absolute affinity values, rather than merely comparative values provided by DSM-only models. The ability to quantify absolute binding affinities is essential for drug efficacy study.

- By synthesizing the strengths of both DSM and MSE, DualBind also exhibits better generalization capability compared to MSE-only models, leading to improved prediction performance.

- Another significant advantage of DualBind is its ability to effectively utilize both labeled and unlabeled data simultaneously. Unlike MSE-only methods which are heavily dependent on the presence of labeled data, DualBind's dual-loss framework significantly reduces this dependency. This flexibility is particularly valuable in scenarios where labeled data are scarce. To be specific, for data samples where affinity labels are available, the model leverages a combination of MSE and DSM losses, as formulated in Eq. (7). For unlabeled crystal structures, we can ignore the upper branch in Figure 2 and only use the DSM loss to train the model. The effectiveness of this approach is validated through experiments detailed in Section 3. We demonstrate that DualBind, by using additional unlabeled data, significantly outperforms MSE-only models, when both are trained with the same quantity of labeled data. This shows DualBind's unique capability to harness the full potential of both labeled and unlabeled data.

## 3. Experiments

In this section, we evaluate DualBind empirically to demonstrate the previously outlined advantages.

**Dataset.** Following Jin et al. (2023b), we use the refined subset of PDBbind v2020 (Su et al., 2018) as our training set, which has 4643 protein-ligand complexes. The validation set curated by Stärk et al. (2022) has 357 complexes. For testing, we employ the widely used CASF-2016 benchmark (Su et al., 2018), which consists of 285 high-quality protein-ligand complexes known for their reliable binding affinity measurements. We have excluded any training samples that overlap in ligands with those in the validation and test sets, resulting in a slightly smaller training set size for DualBind (4643 complexes) compared to DSMBind (4806 complexes).

**Setup.** The noise scale $\sigma$ in Eq. (2) is uniformly sampled from the interval $[0.1, 1]$ during training. We use $\lambda = 2$ as the weight for DSM loss. We measure the Pearson cor-

Table 1. **Performance comparison on the CASF-2016 benchmark.** ↑ (↓) represents that a higher (lower) value denotes better performance. "N/A" in the RMSE column represents that the metric is not applicable, as certain methods yield only comparative values instead of absolute affinity values. The results for scoring functions are sourced from Su et al. (2018), results for K_DEEP from Kwon et al. (2020), and results for other methods from their respective publications.

| Method | Affinity labels | $R_p\uparrow$ | RMSE↓ | $\rho\uparrow$ |
|---|---|---|---|---|
| Glide-XP | ✗ | 0.467 | 1.95 | - |
| Glide-SP | ✗ | 0.513 | 1.89 | - |
| Autodock Vina | ✗ | 0.604 | 1.73 | - |
| DSMBind (Gaussian) | ✗ | 0.638 | N/A | - |
| DSMBind (SE(3)) | ✗ | 0.656 | N/A | - |
| K_DEEP | ✓ | 0.738 | 1.462 | - |
| PIGNet | ✓ | 0.749 | - | - |
| DualBind | ✓ | **0.757**±0.006 | **1.461**±0.013 | **0.742**±0.008 |
| MSE-only | ✓ | 0.749±0.008 | 1.491±0.017 | 0.736±0.006 |
| DSM-only | ✗ | 0.646±0.005 | N/A | 0.652±0.007 |

relation coefficient ($R_p$) to assess the linear relationship between the predicted and actual binding affinity. The root mean square error (RMSE) is used to quantify the average magnitude of the prediction errors, offering a direct measure of prediction error across all test cases. Additionally, the Spearman correlation coefficient ($\rho$) is reported to assess the rank-order accuracy of our predictions, reflecting how well the model can predict the relative order of binding affinities in the dataset. We further assess the impact of each component in DualBind by evaluating two ablated models: one retaining only the MSE loss and the other only the DSM loss, referred to as MSE-only and DSM-only, respectively. For DualBind and its ablated models, we report the average performance across three random runs. The model checkpoints are selected based on achieving the highest Pearson correlation coefficient on the validation set.

**Baselines.** We consider several empirical scoring functions as baselines, including Glide-XP and Glide-SP (Friesner et al., 2004), along with AutoDock Vina (Trott & Olson, 2010; Eberhardt et al., 2021). Additionally, we compare against two versions of DSMBind that use Gaussian noise and SE(3) noise respectively (Jin et al., 2023b). In the realm of supervised machine learning methods, we include K_DEEP (Jiménez et al., 2018) and PIGNet (Moon et al., 2022), which use training sets similar in size to ours. It is important to note that direct comparisons across all models are challenging due to differences in dataset sizes and splits used for training. We emphasize that the primary aim of our study is not to outperform all supervised methods in the literature but to demonstrate the effectiveness of our dual-loss approach, which is designed to be model-agnostic.

**Results.** As summarized in Table 1, DualBind achieves a Pearson correlation of 0.757, significantly outperforming scoring functions and DSM-based methods. Unlike DSM-only models, which do not provide absolute affinity predic-

tions (as indicated by "N/A" for RMSE), DualBind achieves an RMSE of 1.461, affirming its capability to deliver accurate absolute affinity values. Furthermore, as demonstrated in Figure 3, the rank fit of our DualBind model on training complexes significantly surpasses that of DSM-only models (Figure 1 (b)), achieving a Spearman correlation of 0.820 compared to 0.522. This highlights DualBind's enhanced capability in accurately ranking protein-ligand interactions, overcoming the limitations posed by the unrealistic Boltzmann distribution assumption in DSM-only models. Hence, on the test set, DualBind continues to exhibit superior performance with a Spearman correlation of 0.742, compared to 0.652 for DSM-only models. Additionally, when compared to existing supervised models and our MSE-only variant, DualBind exhibits enhanced linear correlation, reduced RMSE, and improved ranking power. These results validate the key advantages outlined in Section 2.

**Experiment on Dual-Bind's Flexible Data Use Strategy.** As outlined in Section 2, because of the dual-loss framework, DualBind can flexibly employ the DSM loss for unlabeled data while using a combination of DSM and MSE losses for labeled data. This allows for the simultaneous training on both labeled and unlabeled datasets. Thus, in this experiment, we

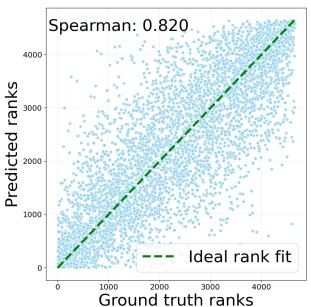

Figure 3. **Rank fit of DualBind on training complexes.**

employ labels from only 50% of the training samples (*i.e.*, 2321 complexes) to train both the MSE-only model and DualBind. For DualBind, we additionally leverage the remaining 50% samples without using their affinity labels.

*Table 2.* **Experimental results on DualBind's flexible data use strategy.** In this comparison, DualBind and the MSE-only model use the same amount of labeled data. DualBind enhances the performance by additionally using unlabeled data, leveraging its unique data use capability. $\uparrow$ ($\downarrow$) represents that a higher (lower) value denotes better performance.

| Method | #Labeled | #Unlabeled | $R_p\uparrow$ | RMSE$\downarrow$ | $\rho\uparrow$ |
|---|---|---|---|---|---|
| MSE-only | 2321 | ✗ | $0.664_{\pm0.037}$ | $1.694_{\pm0.086}$ | $0.666_{\pm0.028}$ |
| DualBind | 2321 | 2321 | $\mathbf{0.731}_{\pm0.007}$ | $\mathbf{1.684}_{\pm0.087}$ | $\mathbf{0.732}_{\pm0.006}$ |
| MSE-only | 4643 | ✗ | $0.749_{\pm0.008}$ | $1.491_{\pm0.017}$ | $0.736_{\pm0.006}$ |

Specifically, for DualBind, we randomly sampled data from both the labeled and unlabeled subsets to compute their respective loss values, followed by back-propagation to train the model. The goal of this experimental setup is to effectively demonstrate DualBind's unique capacity to utilize both labeled data and unlabeled data, a functionality not available in MSE-only models.

The average results over three random runs are reported in Table 2. Notably, DualBind significantly outperforms the MSE-only model when using an equivalent amount of labeled data, achieving a Pearson correlation of 0.731 compared to 0.664. This improvement shows DualBind's effective use of additional unlabeled data. For comparison, we also include results from an MSE-only model trained with 100% labeled data. Impressively, DualBind, utilizing only half labeled data and half unlabeled data, achieves results nearly comparable to the MSE-only model trained on the full labeled dataset, in terms of both linear and ranking correlation coefficients. It clearly demonstrates DualBind's superior ability to enhance model performance through its unique data use strategy. This strategic use of data resources also broadens the practical applicability of DualBind, especially in scenarios where acquiring extensive labeled data is challenging.

## 4. Conclusion

In this study, we propose DualBind, a novel dual-loss framework which can perform accurate absolute binding affinity prediction. Our evaluations on standard benchmark demonstrate that DualBind consistently outperforms both baseline models and its ablated variants. A key feature of DualBind is its innovative strategy of integrating unlabeled data with labeled examples, which has shown considerable benefits in our initial experiments. Looking ahead, we plan to expand our research into hybrid training approaches and explore pretraining techniques around the DualBind framework.

## Acknowledgements

We thank the entire NVIDIA BioNeMo team for the discussions and support throughout this research.

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
