# OpenReview forum: "DualBind: A Dual-Loss Framework for Protein-Ligand Binding Affinity Prediction"
_ICML.cc/2024/Workshop/ML4LMS — ML4LMS Poster_

### Official Review · Reviewer_iK9y · 2024-06-03
**Review of short paper**

**Rating:** 6
**Confidence:** 4

**Review:**

It is unclear what the authors mean that the training data follows a Boltzmann distribution in the introduction. From a statistical mechanics point of view this argument and also the explanation in 2.1 are rather crude. The authors likely just want to say that the sample of available protein ligand complexes has certain biases wherefore models might overfit or cannot be trained properly to e.g predict low affinity complexes as evinced by Figure 1.

I don't think its fair to say that " DualBind is not strictly dependent on the availability of labeled data". In the limit of no labeled data the model just is the DSM-only model.

---

### Official Review · Reviewer_2oS7 · 2024-06-10
**A novel method integrating supervised and unsupervised training for ligand affinity prediction.**

**Rating:** 6
**Confidence:** 4

**Review:**

# Summary:

The paper presents DualBind, a novel dual-loss framework that combines MSE loss with Denoising Score Matching (DSM) loss to enhance protein-ligand binding affinity predictions. This model aims to improve the accuracy of predictions by leveraging the strengths of unsupervised pretraining with perturbated structure dataset and supervised training with true affinity labels. DualBind is evaluated against various benchmarks and demonstrates superior performance in predicting binding affinities and ranking protein-ligand interactions. The main contribution is the combination of two types of loss - unsupervised DSM and supervised MSE. This overcomes the shortcoming of Jin et at 2023 that learned energies cannot be mapped to true affinity labels.

# Strengths and weaknesses:

**Originality**: The idea of using both supervised and unsupervised data for affinity prediction is novel, though the technical difficulty may not be particularly high. However, the two components are previously established concepts already. Although the authors claim that the assumption of Boltzmann distribution for training energy-based models is eliminated here, the framework used to train the model still relies on this assumption. Adding another dimension - supervised training - to the loss does not reduce the dependence on Boltzmann distribution assumption. Method section 2.1 is questionable - DSM model's struggle with accuracy does not necessarily justify the flaw of this assumption.

**Quality**: Data presented in this paper is well-documented with statistical analysis. A comprehensive benchmark of other methods is provided. The performance of proposed method is showcased from various problem settings.

**Clarity**: The paper is well-written and easy to follow. Figures and tables are well-formatted. It would benefit to have a sketch of deep learning framework used in the paper.

**Significance**: Even though there is a performance gain shown by various metrics, it is not significant. Supervised training with affinity labels can already have superior scores than DSMBind model. The gain from dual loss is on the same order of magnitude of SDs, which is not particularly impressive. However, the combination of supervised and unsupervised training for this problem is an interesting direction to work on.